# Determination of the MiRNAs Related to Bean Pyralid Larvae Resistance in Soybean Using Small RNA and Transcriptome Sequencing

**DOI:** 10.3390/ijms20122966

**Published:** 2019-06-18

**Authors:** Weiying Zeng, Zudong Sun, Zhenguang Lai, Shouzhen Yang, Huaizhu Chen, Xinghai Yang, Jiangrong Tao, Xiangmin Tang

**Affiliations:** 1Cash Crops Research Institute, Guangxi Academy of Agricultural Sciences, Nanning 530007, China; zengweiying_1981@163.com (W.Z.); laizhenguang@126.com (Z.L.); 13367811306@189.cn (S.Y.); chhuaizhu@sina.com (H.C.); tjrwring1995@163.com (J.T.); TXM906@126.com (X.T.); 2Rice Research Institute, Guangxi Academy of Agricultural Sciences, Nanning 530007, China; yangxinghai514@163.com

**Keywords:** soybean, bean pyralid, miRNA, target genes

## Abstract

Soybean is one of the most important oil crops in the world. Bean pyralid is a major leaf-feeding insect of soybean. In order to screen out the functional genes and regulatory pathways related to the resistance for bean pyralid larvae, the small RNA and transcriptome sequencing were performed based on the highly resistant material (Gantai-2-2) and highly susceptible material (Wan 82-178) of soybean. The results showed that, when comparing 48 h feeding with 0 h feeding, 55 differentially expressed miRNAs were identified in Gantai-2-2 and 58 differentially expressed miRNAs were identified in Wan82-178. When comparing Gantai-2-2 with Wan82-178, 77 differentially expressed miRNAs were identified at 0 h feeding, and 70 differentially expressed miRNAs were identified at 48 h feeding. The pathway analysis of the predicted target genes revealed that the plant hormone signal transduction, RNA transport, protein processing in the endoplasmic reticulum, zeatin biosynthesis, ubiquinone and other terpenoid-quinone biosynthesis, and isoquinoline alkaloid biosynthesis may play important roles in soybean’s defense against the stress caused by bean pyralid larvae. According to conjoint analysis of the miRNA/mRNA, a total of 20 differentially expressed miRNAs were negatively correlated with 26 differentially expressed target genes. The qRT-PCR analysis verified that the small RNA sequencing results were credible. According to the analyses of the differentially expressed miRNAs, we speculated that miRNAs are more likely to play key roles in the resistance to insects. Gma-miR156q, Gma-miR166u, Gma-miR166b, Gma-miR166j-3p, Gma-miR319d, Gma-miR394a-3p, Gma-miR396e, and so on—as well as their negatively regulated differentially expressed target genes—may be involved in the regulation of soybean resistance to bean pyralid larvae. These results laid a foundation for further in-depth research regarding the action mechanisms of insect resistance.

## 1. Introduction

MicroRNA (miRNA) is a type of non-coding RNA (small RNA) approximately 20–24 nt in length, which widely exists in both higher and lower organisms. It is an important regulatory factor for gene expression at the transcriptional and post-transcriptional levels [1]. The functions of miRNA in plant development, environmental adaptation, stress resistance, and so on, have been confirmed, it is even considered that miRNA is a new level of gene regulation [2,3,4,5]. Soybean is subjected to biotic and abiotic stresses during its growth stages. It is known that soybean miRNA also plays an important role in regulatory gene expressions during various stresses. For example, 10 miRNAs were associated with the infection and resistance of *Phytophthora sojae*, and over-expressions of these differential miRNAs could impact the pathogenicity of *Phytophthora sojae*, Gma-miR393 and Gma-miR166 may be involved in the basic defense against *Phytophthora sojae* [6,7]. Gma-miR160, Gma-miR393, and Gma-miR1510 responded to SMV infections [8]. There were 101 miRNAs in 40 miRNA families involved in the infection defense against SCN; Gma-miR393, Gma-miR1507, Gma-miR1510, Gma-miR1515, Gma-miR171, and Gma-miR2118 were able to produce phase RNA in order to respond to SCN infections [9,10]. There were 126 miRNAs related to phosphorus stress, of which 112 were simultaneously expressed in the roots and leaves [11]. A total of 27 known miRNAs, 16 conserved miRNAs and 12 new miRNAs responded to phosphate deficiency in roots, 34 known miRNAs, 14 conserved miRNAs, and 7 new miRNAs responded to phosphate deficiency in seedlings [12]. It was found that 26 miRNAs responded to cadmium stress, of which 12 were expressed only in cadmium susceptible varieties, 5 in cadmium resistant varieties, and 9 in both varieties [13]. In addition, 30 miRNAs were found to have responded to aluminum stress [14]. Therefore, as demonstrated from the results of the aforementioned research regarding the regulation functions of soybean miRNA, a deeper understanding of the mechanisms of crop resistance has been achieved.

Bean pyralid (*Lamprosema indicata* Fabricius) is an important leaf-feeding pest of soybean, it is widely distributed throughout the world and is found in Korea, Japan, China, India, the Americas, and Africa [15]. It can only feed and reproduce on soybean, so soybean is its only food source. Bean pyralid larvae begin to roll the leaves after the third instar and then lie in them. It feeds on the leaf tissues, consequently soybean cannot perform normal photosynthesis and lose nutrients, preventing the normal growth, so that the flowers and pods fall off and the yield is reduced. These serious harmful effects can cause sharp decreases in yield or even total crop failure of soybean [16]. In southern China, four to five generations can occur in one year. In years with serious damage, only veins and petioles have been left on the blades after leaves have been consumed, generally, the yield is reduced between 15% and 20%, and when the infestation is severe, the yield can be reduced more than 30% [17]. We have been studying on soybean resistance to bean pyralid for a long time, including resistance resource mining, resistance identification, resistance inheritance, physiology, biochemistry, etc. For example, after some years of resistance identification, the results showed that Gantai-2-2 from Jiangsu was highly resistant, Wan82-178 from Anhui was highly susceptible, and resistance to bean pyralid is stable. The soybean resistance to bean pyralid was evaluated by the density of package and insect mouth [17]. Long et al. found that the pupation rate, generation survival rate, and egg hatching rate of adult bean pyralid were different in the different genotypes of soybean varieties. The pupation rate and generational survival rate of bean pyralid were the lowest among Gantai-2-2, the results showed that Gantai-2-2 could significantly inhibit the growth and oviposition of bean pyralid [18].

Therefore, further study regarding the regulations of miRNAs under bean pyralid larvae stress will deepen our understanding of the molecular mechanisms of soybean insect resistance and allow better use of miRNAs to enhance soybean resistance. The leaves of Gantai-2-2 and Wan82-178 were used to identify the differentially expressed miRNAs and their target genes related to the regulation of the resistance to bean pyralid larvae through high-throughput sequencing and bioinformatics analyses. Then study could be conducted on the metabolic pathways and molecular networks of the differentially expressed miRNAs and their involved target genes. The purpose of this study was to explore the functions of the specific miRNA/target genes resistance to bean pyralid stress in soybean, and to develop a preliminary understanding of the pathways of soybean resistance to bean pyralid mediated by miRNA/target genes. Functional verification will be carried out by screening targeted miRNA/target genes related to insect resistance, so as to further analyze the insect resistance regulation mechanism involved by miRNA/target genes and create new insect-resistant transgenic materials. It will open up new ideas for further clarifying the molecular mechanism of soybean resistance to bean pyralid, and also provide a breakthrough for the development of specific insect-resistant elements to find the level of target gene regulation. These results will lay a solid theoretical foundation for soybean insect-resistant molecular breeding.

## 2. Results

### 2.1. Analysis of the Small RNA Sequencing Data

For the purpose of understanding the roles of miRNAs in the regulation of soybean resistance to bean pyralid larvae, eight small RNA libraries were constructed for Gantai-2-2 (HRK) and Wan82-178 (HSK) before and after bean pyralid larvae feeding. Then, high-throughput sequencing was performed using an Illumina sequencing technique. More than 15 million raw reads were obtained in the eight small RNA libraries, respectively (Table 1). Then, following the filtering of the low-quality sequences; 3′ and 5′ adapter sequences; sequences without insert fragments; sequences with overly long insert fragments; polyA sequences; and fragment sequences <18 nt, more than 14 million clean reads were obtained, ranging from 14335242 to 20230796. Among these, 12999357 to 18078009 small RNA sequences were matched into soybean genome, respectively (Table 1).

The lengths of these small RNAs sequences in leaves were analyzed statistically in accordance with the sequencing results (Figure 1). The results showed that the lengths of the small RNAs ranged 18–32 nt, and majority of the small RNAs in the eight libraries were 20–24 nt. Also, the distribution peaks were concentrated in 20 nt, 21 nt, and 24 nt.

The classified annotations of the small RNAs showed that the number of unannotated small RNAs in the eight samples was much larger than the number of annotations in both quantity and species (Table 1). In terms of quantity, the proportion of the miRNAs was found to be larger than that of the other types of non-coding RNAs. However, the number of types was smaller than that of the other types of non-coding RNAs. In addition, the ratios of the rRNAs were determined to be 1.26%, 2.37%, 2.76%, 1.46%, 1.59%, 1.74%, 1.68%, and 1.87%, all of which were much lower than 60% [19]. These results indicated that the samples were of good quality, and the obtained data were reliable.

### 2.2. Identification and Conservation Analyses of the MiRNAs

All mappable small RNA sequences were compared with the known soybean miRNAs in the miRbase database (miRbase 21.0, http://www.mirbase.org/), 428 mature miRNAs belonging to 192 miRNA families was completed. Among these known families of miRNAs, MIR166 contained 18 members; followed by MIR156 with 16 members, MIR319 with 16 members, MIR171 with 15 members, MIR396 with 14 members, and MIR172 with 11 members. However, the majority of the miRNA families were found to have fewer than 10 members. Most of the miRNAs (51.84%) were conserved in the different plant species and played important and conserved roles in plant development, including leaf tissue development. In contrast, the non-conserved miRNAs were observed to only have species-specific characteristics in plant development. Zhang et al. demonstrated that the miRNA families classified as highly-, moderately-, lowly-, and non-conserved miRNAs found in more than 10, 5–9, 2–4, and only 1 plant species [20]. There were 23 highly conserved; 3 moderately conserved; 20 lowly conserved; and 146 non-conserved miRNAs families have been identified in the eight samples (Table 2).

Also, according to the prediction results, 61 putative novel miRNAs (Appendix A) in 61 new families were identified. The sequence lengths of these mature miRNAs were found to range from 19 to 30 nt. The precursors of these miRNAs were identified by MIREAP. The length distributions were found to range from 69 to 910 nt. The first base in the sequence of the majority of the mature novel miRNA was U, which was consistent with the results of the previously reported soybean miRNA.

### 2.3. Analyses of the Differentially Expressed MiRNAs

The expression quantities of some miRNAs were found to change significantly during the process of bean pyralid larvae stress. Therefore, in order to determine the miRNAs which were associated with soybean resistance to bean pyralid larvae, analyses were made of the different expressions of soybean miRNAs in two control groups (HRK48/HRK0 and HSK48/HSK0). The results showed that 55 differentially expressed miRNAs (52 known miRNAs and 3 novel miRNAs) were identified in Gantai-2-2 when comparing 48 h feeding with 0 h feeding, of which 28 miRNAs were upregulated and 27 were downregulated. A total of 58 differentially expressed miRNAs (45 known miRNAs and 13 novel miRNAs) were identified in Wan82-178, of which 25 miRNAs were upregulated and 33 were downregulated (Appendix A). In accordance with the expressions of the miRNAs, the number of the upregulated miRNAs was not found to be significantly different from that of the downregulated miRNAs when soybean was stressed by bean pyralid larvae. These findings suggested that some of the miRNAs in soybean had become activated, and some of the miRNAs were inhibited after stimulation. It was speculated that the significant or very significant differentially expressed miRNAs may have been responding to the stress caused by bean pyralid larvae through the upregulated or downregulated differential expressions.

In order to screen the constitutive defense genes, miRNAs were identified by analyzing the expression of the miRNAs in the resistant and susceptible materials without bean pyralid larvae feeding (HRK0/HSK0). It was found that, a total of 77 differentially expressed miRNAs (69 known miRNAs and 8 novel miRNAs) were identified when comparing Gantai-2-2 with Wan82-178, of which 47 miRNAs were upregulated and 30 were downregulated. Among these, Gma-miR4352b and Gma-miR5040 were expressed only in Gantai-2-2 and had almost no expression in Wan82-178. These results indicated that these two miRNAs were specifically expressed in the insect-resistant material. Also, in accordance with the genotype-specific expressions of the miRNAs without insect stress, it was speculated that some miRNAs would also be preferentially expressed in certain material after being invaded by bean pyralid larvae. Therefore, the expression of the miRNAs in the resistant and susceptible materials were also analyzed at 48 h (HRK48/HSK48). A total of 70 differentially expressed miRNAs (60 known miRNAs and 10 novel miRNAs) were identified when comparing Gantai-2-2 with Wan82-178, of which 42 miRNAs were upregulated and 28 were downregulated (Appendix A). It was speculated that these differentially expressed miRNAs regulated soybean resistance to the infringement of bean pyralid larvae by upregulating or downregulating their expressions.

There were three types of differentially expressed miRNAs in the two materials. The first type included “differentially expressed miRNAs with non-bean pyralid-induced genotype”. There were 77 miRNAs identified in total. This type of miRNAs were the “differentially expressed miRNAs identified in Gantai-2-2 compared to Wan82-178 before bean pyralid feeding”, in which, 17 miRNAs were always upregulated and 12 were always downregulated at 0 h and 48 h, respectively. In addition, 3 miRNAs were found to be upregulated at 0 h but downregulated at 48 h. Furthermore, 5 miRNAs were downregulated at 0 h but upregulated at 48 h. The other 40 miRNAs displayed no regulation differences. The second type included the “bean pyralid-induced differentially expressed miRNAs which appeared in both the insect-resistant and insect-susceptible materials”. A total of 24 miRNAs were determined to belong to this type, of which 10 miRNAs were upregulated and 6 were downregulated in both materials; 3 miRNAs were downregulated in Gantai-2-2 but upregulated in Wan82-178; and 5 miRNAs were upregulated in Gantai-2-2 but downregulated in Wan82-178. The third type of miRNAs were the “bean pyralid-induced genotype differentially expressed miRNAs”. There were 65 miRNAs in this type, such miRNAs which were only expressed in the resistant or susceptible material. Among these, 31 miRNAs were expressed only in Gantai-2-2 and 34 miRNAs in Wan82-178.

### 2.4. Cluster Analyses of the Differentially Expressed MiRNAs

The cluster analyses results of the differentially expressed miRNAs in the four control groups showed that these groups were divided into two modules (Figure 2). In the first module, HRK48/HRK0 and HRK48/HSK48 were found to have similar overall expression patterns. In the second module, HSK48/HSK0 and HRK0/HSK0 also had similar overall expression patterns. These similarities were mainly manifested in simultaneous upregulations or downregulations. Moreover, these results indicated that the insect-induced miRNAs of the two genotype materials following the feeding stress caused by bean pyralid larvae were able to respond to the insect stress.

### 2.5. Analysis of the qRT-PCR of the Differential MiRNAs

The differentially expressed miRNAs were verified using a stem-loop qRT-PCR method. During the verification, 10 differentially expressed miRNAs were randomly selected. It was found that 8 miRNAs had identical sequencing results in the four control groups (Figure 3). It was observed that Gma-miR159e-5p was downregulated by small RNA sequencing but upregulated by the stem-loop qRT-PCR in HRK48/HSK48; however, it was consistent with the sequencing results in the other three control groups. Gma-miR156a-5p was downregulated by small RNA sequencing but upregulated by the stem-loop qRT-PCR in HRK0/HSK0, yet it was consistent with the sequencing results in the other three control groups. Therefore, the miRNAs with inconsistent expression trends required further verification.

### 2.6. Target Gene Prediction and the Functional Analyses of the Differentially Expressed MiRNAs

The roles of miRNAs mainly include the regulation of plant-related physiological processes through the splicing or translation inhibition of target genes. Therefore, target gene predictions are important links in understanding the biological functions of miRNAs [21]. The target gene predictions of differentially expressed miRNAs can directly capture some of the genes which are affected by the differentially expressed miRNAs (Appendix A). Furthermore, target gene predictions can help to further understand the regulatory roles of miRNAs in soybean resistance against bean pyralid larvae, so the target gene predictions were made on the differentially expressed miRNAs in the four control groups using prediction software. The results showed that, when comparing 48 h feeding with 0 h feeding, the number of predicted target genes and predicted target gene loci in Gantai-2-2 were 608 and 748; and in Wan82-178 were 704 and 928, respectively. When comparing Gantai-2-2 with Wan82-178 at 0 h feeding, the number of predicted target genes and predicted target gene loci were 1010 and 1133, respectively. When comparing Gantai-2-2 with Wan82-178 at 48 h feeding, the number of predicted target genes and predicted target gene loci were 637 and 761, respectively. It was observed that the regulation relationships between the miRNAs and their target genes were not always one-to-one, at times, one miRNA would regulate multiple target genes. Meanwhile, at other times, one target gene was regulated by multiple miRNAs. Generally speaking, the target genes of the same miRNA belonged to the same gene family [22].

GO (gene ontology) function analysis was conducted on the predicted target genes, in order to preliminarily understand the possible biological functions of the differentially expressed miRNAs and facilitate the screening of the candidate miRNAs. The results showed that, when comparing 48 h feeding with 0 h feeding, 442 (72.70%) of 608 target genes predicted in Gantai-2-2 were annotated into 42 functional groups (Figure 4A), including 16 biological processes, 14 cellular components, and 12 molecular functions, and 527 (74.86%) of 704 target genes predicted in Wan82-178 were annotated in 41 functional groups (Figure 4B), including 17 biological processes, 13 cellular components, and 11 molecular functions. When comparing Gantai-2-2 with Wan82-178 at 0 h feeding, 794 (78.4%) of the 1010 target genes predicted were annotated in 41 functional groups (Figure 4C), including 17 biological processes, 12 cellular components, and 12 molecular functions. When comparing Gantai-2-2 with Wan82-178 at 48 h feeding, 492 (77.24%) of 637 target genes predicted were annotated in 41 functional groups (Figure 4D), including 17 biological processes, 13 cellular components, and 11 molecular functions. From the enrichment results of the four control groups, it can be seen that the predicted target gene functions were similar among the different control groups. For example, during the biological process, the target genes were mainly involved in the cell, metabolic, single-organism, and biological regulation processes. In the cell component, the target genes were mainly concentrated in the cells, cell parts, organelle, membranes, and membrane parts. In the molecular function, the target genes were mainly involved in the functions of binding and catalytic activities. The role of the differentially expressed miRNAs could be preliminarily understood through the analysis of the aforementioned functions in order to facilitate the screening of the miRNAs with specific functions.

In organisms, different genes coordinate with each other to perform specific biological functions. Pathway-based analysis assists in furthering our understanding of the biological functions of genes. It has been found that significant pathway enrichment analysis can determine the main biochemical metabolic pathways and signal transduction pathways in which the predicted target genes participate. The results revealed that, when comparing 48 h feeding with 0 h feeding, 478 (78.61%) of 608 target genes predicted in Gantai-2-2 were enriched into 75 pathways, including 11 significantly enriched pathways, the main pathways were RNA transport (25, 5.23%), protein processing in endoplasmic reticulum (23, 4.81%), zeatin biosynthesis (8, 1.67%), ubiquinone and other terpenoid-quinone biosynthesis (8, 1.67%). In addition, 551 (78.27%) of 704 target genes predicted in Wan82-178 were enriched in 79 pathways, including 6 significantly enriched pathways, these pathways mainly included plant hormone signal transduction (111, 20.15%), arginine and proline metabolism (13, 2.63%), zeatin biosynthesis (8, 1.45%), ubiquinone and other terpenoid-quinone biosynthesis (8, 1.45%). When comparing Gantai-2-2 with Wan82-178 at 0 h feeding, 864 (85.54%) of 1010 predicted target genes were enriched into 91 pathways, including 12 significantly enriched pathways, these pathways mainly included plant hormone signal transduction (105, 12.15%), RNA transport (32, 3.70%), arginine and proline metabolism (17, 1.97%). When comparing Gantai-2-2 with Wan82-178 at 48 h feeding, 552 (86.66%) of 637 predicted target genes were enriched in 78 pathways, including 15 significantly enriched pathways, these pathways mainly included plant hormone signal transduction (30, 5.75%), RNA transport (24, 4.60%), isoquinoline alkaloid biosynthesis (18, 3.45%), and tyrosine metabolism (18, 3.45%). It was speculated that the plant hormone signal transduction, RNA transport, protein processing in endoplasmic reticulum, zeatin biosynthesis, ubiquinone and other terpenoid-quinone biosynthesis, and isoquinoline alkaloid biosynthesis played important roles in the defense process of soybean against the stress caused by bean pyralid larvae. Therefore, it was speculated that the miRNAs displayed the ability to improve the plants resistance to insects by regulating the expressions of related genes in the aforementioned metabolic pathways when the plants were under biotic stress. These results also confirmed that the mechanism of soybean insect resistance is a very complex process, which involves the interactions of multiple metabolic pathways.

### 2.7. Conjoint Analysis of the Negative Regulations of the MiRNA/mRNA under Bean Pyralid Larvae Stress

The miRNAs in plants are usually completely complementary to the coding sequences of target genes in order to splice and degrade the mRNA, thereby regulating the physiological processes of the plants. Therefore, the accuracy of target gene screening can be improved by using the conjoint analysis of the small RNA sequencing and transcriptome sequencing. Due to the fact that the miRNAs complete post-transcriptional regulations of target genes mainly by suppressing or even silencing the expressions of the target genes, they are generally considered to be negative regulations. A conjoint analysis was made on the differentially expressed genes [23] and the differentially expressed miRNAs. The downregulated target genes were obtained through the upregulated miRNAs. Meanwhile, the upregulated target genes were obtained through the downregulated miRNAs. The results showed that a total of 20 differentially expressed miRNAs were negatively correlated with 26 differentially expressed target genes in the four control group (Table 3). There were 6 differentially expressed miRNAs were negatively correlated with 7 differentially expressed target genes in HRK48/HRK0, 9 differentially expressed miRNAs were negatively correlated with 9 differentially expressed target genes in HSK48/HSK0, 7 differentially expressed miRNAs were negatively correlated with 7 differentially expressed target genes in HRK0/HSK0, 8 differentially expressed miRNAs were correlated with 8 differentially expressed target genes in HRK48/HSK48. It was found that Gma-miR4996 regulated three target genes; Gma-miR166u, Gma-miR1535a, Gma-miR394a-3p, Gma-miR395g, Gma-miR5761a and novel-miR36 regulated two target genes, respectively; and the other 13 miRNAs regulated one target gene, respectively. Meanwhile, Gma-miR166b, Gma-miR166j-3p, and Gma-miR166u were observed to regulate the same one target gene. These results revealed that complex regulatory networks of functional redundancy had occurred in the target genes of the miRNAs related to soybean resistance to bean pyralid larvae. The GO function analysis of the negative regulatory target genes associated with the differentially expressed miRNAs confirmed that these target genes were mainly associated with the responses to stimulus, regulations of transcriptions, and DNA-templates, as well as responses to hormones, cell cycles, activations of MAPKK activities, calcium-transporting ATPase activities, and calciumion transmembrane transport. It was speculated that these miRNAs may play important regulatory roles in soybean resistance to bean pyralid larvae. They may also be involved in the process of resistance to bean pyralid larvae, which can be used for further research examinations.

## 3. Discussion

Under normal physiological conditions, the gene expressions in plants are in a dynamic equilibrium state. However, when stimulated by the outside world, the dynamic balance will be broken, and the expressions of some genes will change accordingly. In recent years, many research results have suggested that miRNAs are indispensable regulators in plant responses to biotic and abiotic stresses. MiRNAs are known to play important roles in the defense and immune responses of plants [4,24]. High-throughput sequencing and bioinformatics analyses were used to initially screen out the miRNAs associated with bean pyralid larvae in soybean. Then, target gene predictions were made on the obtained differentially expressed miRNAs, and the conjoint analysis was conducted in combination with the mRNA. The results showed that some miRNAs had participated in the corresponding regulation process of soybean resistance to bean pyralid larvae. It was possible that when soybean was stimulated by bean pyralid larvae, the organisms resisted the insect pests by inhibiting or inducing miRNAs to activate or inhibit its target gene expressions.

MiR156 is known to be a major factor regulating the transition of plants from juvenile stages to adult stages, and plays an important role in regulating plant growth and development [25]. Its expression tends to decrease with the increased age of a plant, while the plant resistance increases with age [25]. It has been reported that the over-expression of miR156 in maize [26] and *Arabidopsis thaliana* [27] resulted in a longer juvenile period for these plants. Rice resistance to brown plant hopper was enhanced when miR156 was silenced [28]. In addition, previous related studies have indicated that miR156 mainly regulates the SPL (squamosa promoter-binding protein-like) transcription factor [29]. When comparing Gantai-2-2 with Wan82-178 at 0 h feeding, Gma-miR156q was downregulated, and the conjoint analysis showed that Glyma.06G238100.1 (squamosa promoter-binding protein 1-like, SPL) was negatively targeted by Gma-miR156q, we concluded that Gma-miR156q may regulate plant growth and development during insect stress by regulating the SPL transcription factor. In this manner, the plants are able to cope with the insect stress.

It has been determined that miR166 plays an important role in the growth and development of monocotyledons and dicotyledons by regulating the HD-Zip (homodomain-leucine zipper) transcription factors [30,31]. HD-Zip is one of the conserved transcription factors in plants. It is composed of a DNA-homologous domain (HD), and additional Leu zipper (Zip) elements. The HD binds to DNA sequences, and the Zip mediates the formation of protein dimer [32]. Recent studies have shown that miR166 plays a role in plant responses to various environmental stresses, such as miR166 was downregulated in potato and maize under salt stress [33,34], and downregulated in rice during chilling injury [35]. Previous studies have shown that the expressions of subfamily I and II genes of the HD-Zip family transcription factor were induced by drought, high salinity, and chilling injury. These two genes are known to participate in hormone signaling pathways, and regulate plant cell expansion, division, and differentiation by interacting with hormone pathway genes and downstream genes, thereby improving the tolerance levels of plants to stresses [36]. For example, GmHZ1 is a subclass I protein in soybean and plays the role of a transcription activator during the process of mosaic virus infection. Its expression is related to the virus resistance of plants, and it is downregulated in SMV resistant plants [37]. Cabello et al. found that, when arabidopsis was treated with 50.0 mmol/L NaCl for 3 d, the HaHB1 and AtHB13 were increased. Arabidopsis with over-expressions of HaHB1 and AtHB13 displayed good cell membrane stability. Furthermore, the leaf senescence induced by the stress was also delayed, and the transgenic plants showed a higher stress tolerance [38]. HaHB4 has also been confirmed as a drought/ABA inducible gene. In previous studies, arabidopsis with over-expressed HaHB4 genes in *Arabidopsis thaliana* displayed strong drought tolerance, salt tolerance, and herbivore resistance [39,40]. When comparing 0 h feeding with 48 h feeding, Gma-miR166u and Gma-miR166j-3p were downregulated in Wan82-178 and Gantai-2-2, and Gma-miR166b was upregulated when comparing Gantai-2-2 with Wan82-178 at 48 h feeding. The conjoint analysis showed that Gma-miR166u, Gma-miR166b, and Gma-miR166j-3p were negatively correlated with target gene Glyma.07G016700.2 (homeobox-leucine zipper protein ATHB-15-like isoform X4, ATHB-15). These findings led us to the conclusion that Gma-miR166u, Gma-miR166b, and Gma-miR166j-3p may have been involved in regulating soybean resistance to bean pyralid larvae through the target negative regulations of the ATHB-15 transcription factor.

MiR319 regulates the growth and development of leaf by regulating the target TCP transcription factors of family [41]. TCPs directly regulate the LOX2 coding gene which is synthesized in the first-step synthesis of jasmonic acid (JA) [42]. Also, miR319 has the ability to regulate the JA-dependent signaling pathway by regulating the expressions of its target gene TCP, thereby improving the host immunity levels [43]. It has confirmed that miR319 plays an important role in plant reactions to biotic and abiotic stresses. For example, miR319 was involved in plant responses to *Botrytis cinerea* and freezing injury stress [44,45]. When Osa-miRNA319a was transferred to *Agrostis stolonifera*, it was found that the drought resistance and salt tolerance levels were improved in the transgenic *Agrostis stolonifera* [46]. JA signaling plays an important role in anti-insect response, regulating the expression of plant downstream defense response genes, and significantly induced responses of defense systems in plant, thereby effectively reducing pests [47,48,49]. Six Gma-miR319 were upregulated and one Gma-miR319 was downregulated when comparing Gantai-2-2 with Wan82-178 at 0 h feeding, and the target gene predictions indicated that TCP family transcription factors were targeted by Gma-miR319. It is speculated that Wan82-178 uses JA signaling to resist pests, this result is consistent with the results of previous transcriptome studies [23]. In addition, five Gma-miR319 in Gantai-2-2 and two Gma-miR319 in Wan82-178 were downregulated after bean pyralid larvae feeding for 48 h, this would mean that the JA signaling was upregulated. It is speculated that when soybean is under the feeding stress of pests, the activation of JA signaling will be induced, and JA signaling plays an important role in the insect defense of soybean with different genotypes. Therefore, Gma-miR319 may have been involved in the defense responses of soybean against pests stress. However, its specific situation requires further functional verification.

It has been determined that miR394 mainly regulates the richness of the F-box domain proteins [50]. Previous studies have shown that when plants had suffered adversity stresses—such as *B. cinerea*, blue light, drought, and so on—the miR394 were downregulated in tomato [51], longan [52], wild *Ipomoea campanulata* L [53], and *Saccharum spp*. [54]. When Gma-miR394a was transferred into *Arabidopsis thaliana*, it was found that Gma-miR394a had the ability to improve the adaptability to drought stress by reducing the leaf water evaporation rates [55]. When comparing 0 h feeding with 48 h feeding, the Gma-miR394a-3p was downregulated in Gantai-2-2. When comparing Gantai-2-2 with Wan82-178 at 0 h feeding, the Gma-miR394a-3p was upregulated. Then, the conjoint analysis showed that Glyma.14G004300.1 (nudix hydrolase 2-like) and Glyma.05G073500.2 (F-box protein SKIP14-like) were negatively targeted by Gma-miR394a-3p. On the basis of our findings, it was speculated that Gma-miR394a-3p possibly participated in soybean resistance to the stress caused by bean pyralid larvae through the negative target regulations of the F-box protein SKIP14-like and nudix hydrolase 2-like.

It has been found that miR396 mainly regulates the plant growth regulator GRF (growth regulating factor) [56]. The GRF regulation mediated by miRNA396 plays an important role in plant growth and development, and is also involved in varieties of stress responses [57]. For example, miR396a in arabidopsis was observed to be significantly upregulated when it was treated with high salinity, low temperatures, and drought [58]. It was found that the expressions of miR396 had been increased by nearly four times on the seventh day after the cyst nematode was introduced, and the expressions of the corresponding target genes GRF1, GRF3, and GRF8 had decreased during the interactions between arabidopsis and cyst nematode [59]. When Ath-miR396a and Ath-miR396b were over-expressed in *Arabidopsis thaliana*, the development of parasitic nematodes had been restricted in transgenic plants [60]. Furthermore, the drought resistance had been enhanced [61]. When Sp-miR396a was over-expressed in tobacco, the drought, salt and low-temperature resistance of transgenic plants had been enhanced [62]. When comparing Gantai-2-2 with Wan82-178 at 0 h feeding, the Gma-miR396e was upregulated. The results of the target gene predictions indicated that the Gma-miR396e had mainly targeted the GRF3 and GRF5. Then, the conjoint analysis showed that Glyma.13G159700.1 (hypothetical protein GLYMA_13G159700) was negatively targeted by Gma-miR396e. It was speculated that Gma-miR396e could potentially enhance the resistance levels of the highly-resistant material against bean pyralid larvae infestation through the negative target regulation of the GRF genes.

## 4. Methods

### 4.1. Materials, Sample Collection, and Total RNA Extraction

Materials which were used for this experimental were Gantai-2-2 (highly resistant material) [17] and Wan82-178 (highly susceptible material) [17]. The materials were planted in an insect controlled net room of the experimental field of the Guangxi Academy of Agricultural Sciences in July 2017. During the entire growth period of soybean, the pesticides and fertilizers were not sprayed. When soybean had grown to a stage where 10 compound leaves were present, bean pyralid larvae in their fourth instar were insect-inoculated according to a density of five insects per seedling. Sampling was conducted at 0 h and 48 h of the insect inoculation, respectively, and two biological replicates were processed for each. The samples were referred to as HRK0-1, HRK0-2, HRK48-1, HRK48-2, HSK0-1, HSK0-1, HSK48-1, and HSK48-2, respectively. Among these samples, HRK represented Gantai-2-2, and HSK represented Wan82-178. The numbers 0 and 48 represented the processing times, and -1 and -2 represented the first and second replicates, respectively. Five plants were taken from each sample for the mixture. These samples were quickly frozen using liquid nitrogen and stored at −80 °C refrigerated conditions for further use. Total RNA from the leaves of eight samples was extracted using Trizol Reagent (Invitrogen, Carlsbad, CA, USA) according to the manufacturer’s protocol.

### 4.2. Library Construction

The qualified libraries were amplified on cBot in order to generate the clusters on the flow-cell (TruSeq SE Cluster Kit V3-cBot-HS, Illumina, SanDiego, CA, USA). Also, the amplified flow-cell was a sequenced single end on the HiSeq 2000 System (TruSeq SBS KIT-HS V3, Illumina; BGI, Shenzhen, Guangdong Province, China), in which read lengths of 50 are the most common sequencing strategy. Filtering the small RNA: 5–10 μg samples of RNA were used, in which separate RNA segments of different sizes were selected using PAGE gel, and 18–30 nt (14–30 ssRNA Ladder Marker, TAKARA) stripes were selected and recycled. 5′ adaptor ligation: A 5′ adaptor connection system was prepared. Reaction conditions: 20 °C for 6 h, then RNA segments of different sizes were separated by PAGE gel, and 40–60 nt stripes were selected and recycled. 3′ adaptor ligation: A 3′ adaptor connection system was prepared. Reaction conditions: 20 °C for 6 h, then RNA segments of different sizes were separated by PAGE gel, and 60–80 nt stripes were selected and recycled. RT-PCR: A First Strand Master Mix and Super Script II (Invitrogen) reverse transcription were prepared. Reaction conditions: 65 °C for 10 m, 48 °C for 3 m, 42 °C for 1 h, and 70 °C for 15 m. PCR amplification: Several rounds of PCR Primer Cocktail and PCR Mix were performed in order to enrich the cDNA fragments. Reaction conditions: 98 °C for 30 s, 12–15 cycles of 98 °C for 10 s and 72 °C for 15 s, 72 °C for 10 m, and a 4 °C hold. Purification of the PCR products: The PCR products were purified with PAGE gel, approximately 110 bp of the recycled product stripes were selected and recycled, and the recycled products were dissolved in an EB solution. The final library was analyzed in two ways: (1) the average molecule length was determined using an Agilent 2100 bioanalyzer instrument (Agilent, Santa Clara, CA, USA), and (2) the library was quantified using real-time quantitative PCR (qPCR) (ABI StepOnePlus Real-Time PCR System, Foster city, CA, USA).

### 4.3. Data Analysis of the Small RNA

After the sequencing was completed, the raw data was processed, and the main processing steps were as follows: (1) the reads with low sequencing quality were removed, (2) the reads with N proportions higher than 10% were removed (where N denotes that the base information could not be determined), (3) the reads with 5′ junction contamination were removed, (4) the reads without 3′ junction sequences were removed, (5) the reads without insertion fragments were removed, (6) the reads containing polyA were removed, and (7) the reads which were less than 18 nt were removed.

After the above treatments were completed, the data which were obtained were clean reads. Then, the types of small RNA (sRNA) sequences (expressed by ‘unique’), and the number of sequences (expressed by ‘total’) were counted. Also, the length distribution of the small RNA sequences was counted. The expressions and distributions of the sRNA on the genome were analyzed by comparing the sRNA with soybean genome (ftp://ftp.jgi-psf.org/pub/compgen/phytozome/v9.0/Gmax) using AASRA comparison software [63]. Then, by comparing the sRNA with repetitive sequences, repeat-associated sRNA were obtained. Repeat-associated sRNA were obtained by comparing the sRNA with the repeat sequences. The rRNA, scRNA, snoRNA, snRNA, and tRNA in the Genbank (ftp://ftp.ncbi.nlm.nih.gov/genbank/) were selected to annotate the small RNA which had been obtained from the sequencing. The rRNA, scRNA, snoRNA, snoRNA, and tRNA were found and removed as much as possible. At this point, the Rfam database (11.0, http://rfam.janelia.org/) was selected to annotate the small RNA sequences which had been obtained from the sequencing, and the rRNA, scRNA, snoRNA, snRNA, and tRNA were found and removed as much as possible. In order to make each of the unique sRNA have a unique annotation, the sRNAs were traversed and annotated in the following priority order: miRNA > piRNA > snoRNA > Rfam > other sRNAs. The sum of the rRNAs in the classified annotation results could then be used as a quality control standard for a sample.

### 4.4. Small RNA Predictions and Small RNA Expression Quantifications

After removing the non-target sRNA sequences—such as the rRNA, tRNA, snRNA, snoRNA and repetitive sequences—as well as the known miRNAs, the remaining unannotated unique reads were used to predict the new miRNAs. The precursors which were based on the miRNAs formed hairpin secondary structures, dicer restriction enzyme cutting sites, and minimum free energy. RIPmiR [64] was used to predict the new miRNAs (the minimum folding free energy (MFE) of the novel miRNA precursor <0.2 kcal/mol/nt; and the minimum folding free energy index (MFEI) >0.85).

TPM [65] was used to standardize the expression levels of the small RNA, which effectively avoided the impacts of the different sequencing quantities on the quantitative accuracy. Therefore, the standardized data could be directly used for follow-up comparative analysis.

The TPM calculation formula was:
$$\mathrm{TPM}=\frac{C*10^{6}}{N}$$

### 4.5. Target Gene Predictions

The process of determining the potential target genes for the miRNAs was necessary for the subsequent analysis. In order to ensure more accurate results, this study utilized psRobot [66] and TargetFinder [67] software to predict the target genes of miRNA in this study. The default parameters of the target gene prediction software were as follows: psRobot: -gl 17, -p 8, -gn 1; TargetFinder: -c 4, and the union set was taken. The filtering was implemented in combination with the corresponding filtering conditions, such as free energy, score value, and so on.

### 4.6. Screening of the Differentially Expressed MiRNA

RNA sequencing is a known to be a random process, and each sequence is uniformly random from its sample [68]. Therefore, based on this assumption, this study assumed that the expression of each gene (transcript) followed a binomial distribution (or Poisson distribution). Then, using the above model, the DEGseq [69] calculated the differential expression based on the MA-plot [70]. It was assumed that C_1 and C_2 were the total number of reads in the comparison of the two samples, respectively, and obeyed binomial distribution. We define M = (log_2_C_1_ − log_2_C_2_) and A = (log_2_C_1_ + log_2_C_2_/2). It can be proven that under the conditions of random sampling, the distribution of M obeys A = a and accords with the approximate normal distribution. Multiple hypothesis testing and corrections were made for the *p*-value of each gene using *Q*-value. If the selection threshold of |Log_2_FC (Fold Change)| ≥ 1 and the *Q*-value ≤ 0.001, it was considered to be a significantly differentially expressed miRNA.

### 4.7. Hierarchical Cluster Analyses

In accordance with the test results of the differentially expressed miRNAs, the ‘pheatmap’ function of R software was used to carry out a hierarchical clustering analysis. It was determined that multiple groups of differentially expressed miRNAs were clustered at the same time. Therefore, the clustering analysis was performed separately for the intersections and unions of the differentially expressed miRNAs.

### 4.8. Bioinformatics Analyses

GO (GO, http://www.geneontology.org/) and functional enrichment analyses were conducted for all the target genes using TermFinder software (http://www.yeastgenome.org/help/analyze/go-term-finder). Then, all of the target genes were mapped to a pathway in the KEGG (Kyoto Encyclopedia of Genes and Genomes) database (http://www.genome.jp/kegg/pathway.html) using Blast_v2.2.26 software. Also, the *p*-value ≤ 0.05 was used as the threshold for the purpose of judging the significance of the GO and KEGG pathway enrichment analyses.

### 4.9. Quantitative Real Time-PCR (qRT-PCR) Analysis

A RT specific primer (Appendix A) was used for the reverse transcription of the miRNA of each sample (2.5 mM dNTP: 2.0 μL, 10×RT Buffer: 2.0 μL, 1.0 μM RT specific primer: 2.0 μL, Total RNA: 2.0 μg, 10.0 u/μL reverse transcriptase: 1.0 μL, 10.0 u/μL RNA enzyme inhibitor: 1.0 μL, RNA-free enzyme: water up to 20.0 μL), and RT reactions was completed on a PCR amplifier (16 °C for 30 m, 42 °C for 40 m, 85 °C for 5 m). After completing the reactions, the samples were placed on ice for standby use or stored at −20 °C. Fluorescent quantitative PCR amplification was implemented (H_2_O: 6.6 μL, 2×PCR MIX: 8.0 μL, 50 pM/μL up primer: 0.2 μL, 50 pM/μL down primer: 0.2 μL, template (reverse transcription product: for example, cDNA): 1.0 μL. Three parallel experiments were carried out for each sample in the 384 hole plate using an ABI ViiA 7 PCR instrument. The data regarding the CT values in the reactions were collected by the corrected threshold setting. The miRNAs in the real-time fluorescence quantitative PCR method used *mir1520d* as the reference gene [71], and a 2^−ΔΔ*C*t^ method was used for the relative quantification. The means of 2^−ΔΔ*C*t^ were considered significantly different at *p* < 0.5.

## 5. Conclusions

The identification processes and expression analyses of miRNAs before and after soybean suffered stress caused by bean pyralid larvae were implemented, in accordance with the changes of miRNAs in the highly resistant and highly susceptible materials, a total of 132 differentially expressed miRNAs were identified which were confirmed to be related to soybean resistance to bean pyralid larvae. The results of the pathway analysis speculated that predicted target genes of these miRNAs may have formed the soybean insect-resistant regulation network through many ways, such as plant hormone signal transduction, RNA transport, protein processing in endoplasmic reticulum, and zeatin biosynthesis. As a result, soybean was able to effectively defend itself against the insect invasion. A combination of the conjoint analysis of miRNA/mRNA was found that 20 differentially expressed miRNAs were negatively correlated with 26 differentially expressed target genes. According to relevant literature reports, it is speculated that Gma-mirR156q, Gma-miR166u, Gma-miR166b, Gma-miR166j-3p, Gma-miR396e, and their corresponding target genes SPL transcription factors, ATHB-1 and GRF may improve the insect resistance of soybean by regulating plant growth and development, Gma-miR319 and its target gene TCP family transcription factors may induce plant defense system response through the regulation of JA signaling. It was speculated that soybean regulated the production and expressions of some of the miRNAs when undergoing stress caused by bean pyralid larvae. In addition, the expressions of coding genes at the transcriptional and translational levels were observed to be regulated through mRNA molecules targeted by miRNAs, and through the interaction between genes, which eventually led to improved resistance against insect pests. This study presented the first report regarding miRNAs associated with soybean resistance to bean pyralid larvae, which will potentially provide a basis for further understanding of plant resistance mechanisms and genetic improvements in resistant plant varieties.

## Figures and Tables

**Figure 1 ijms-20-02966-f001:**
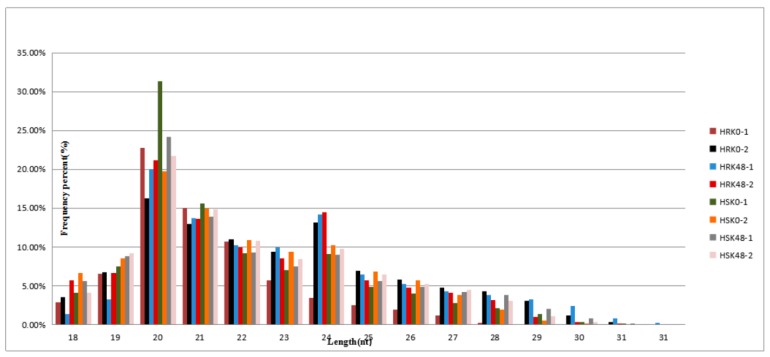
Length distribution of the small RNA.

**Figure 2 ijms-20-02966-f002:**
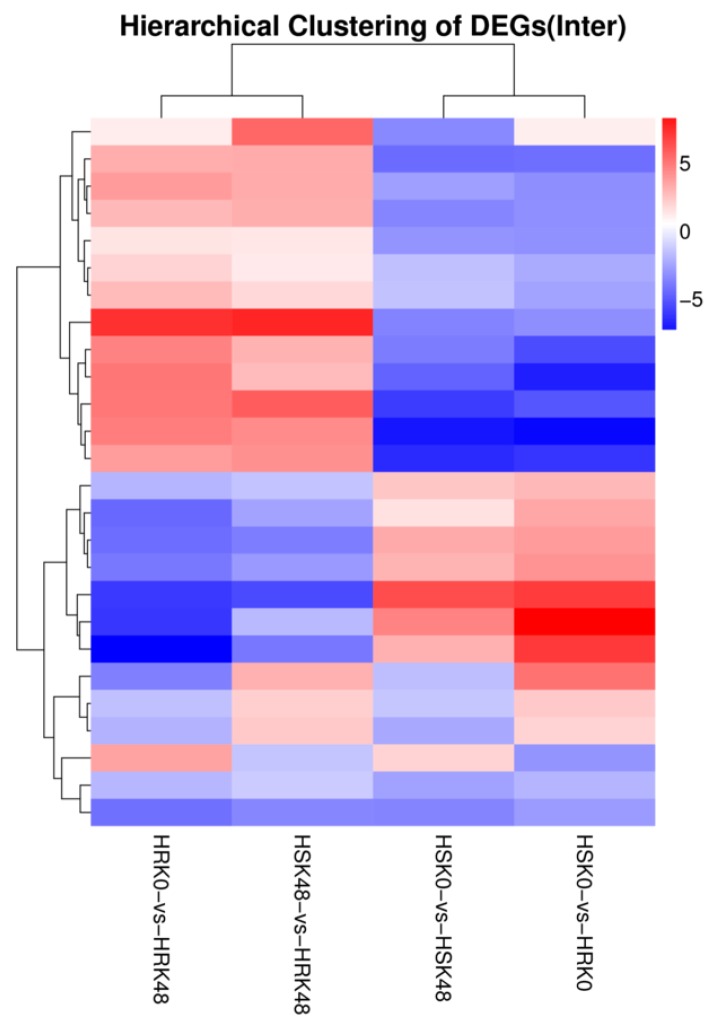
Hierarchical clustering of differentially expressed miRNAs. Note: *x*-axis represents comparing, *y*-axis represents differentially expressed miRNAs. Coloring indicates fold change (upregulated: red, downregulated: blue).

**Figure 3 ijms-20-02966-f003:**
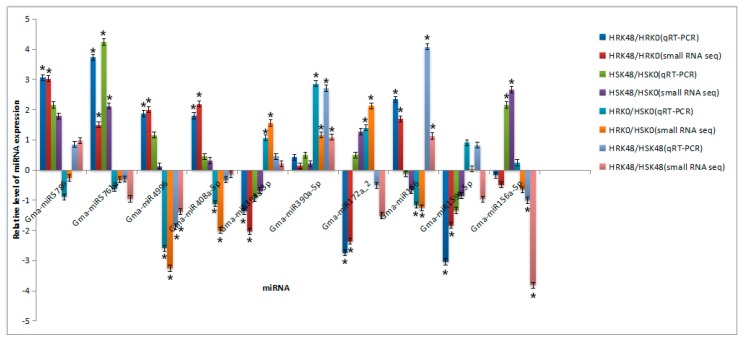
Verification results of differentially expressed miRNAs. Note: “*” representing *p* ≤ 0.01, respectively, which indicated significant difference level.

**Figure 4 ijms-20-02966-f004:**
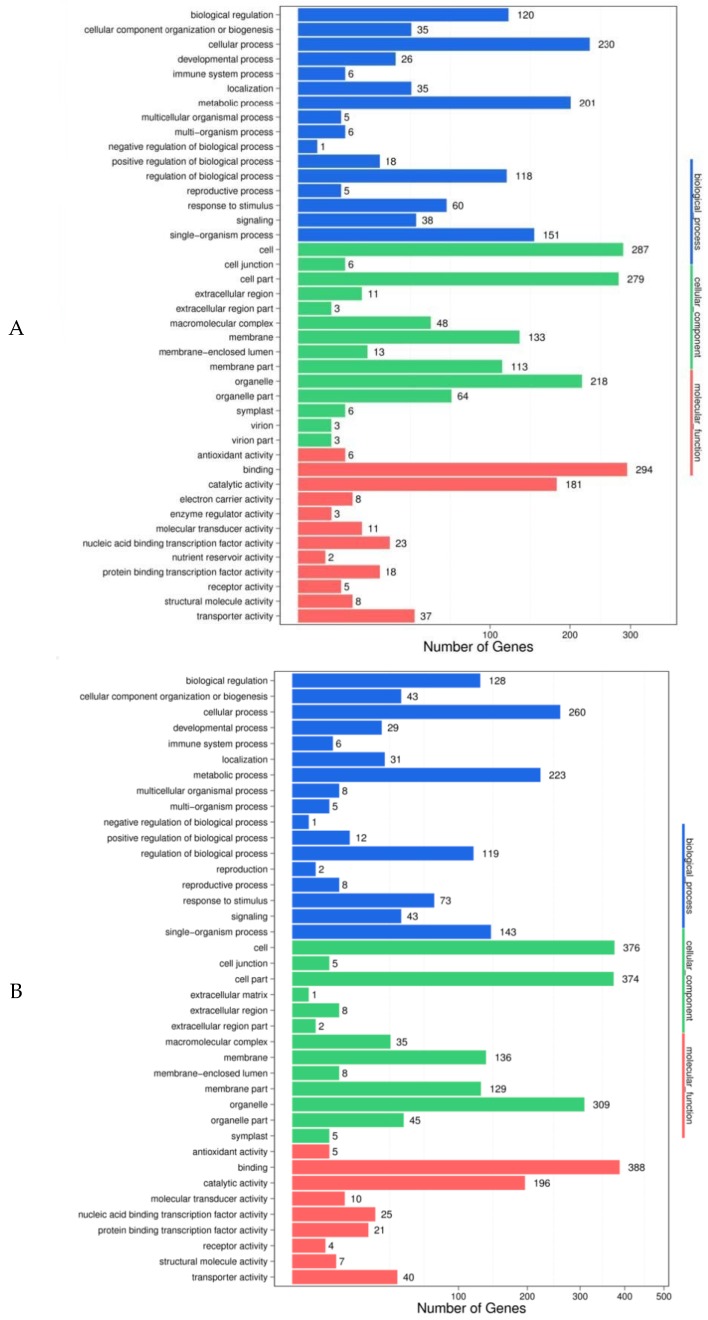
GO function analysis of the DEGs. (**A**) HRK48/HRK0; (**B**) HSK48/HSK0; (**C**) HRK0/HSK0; (**D**) HRK48/HSK48.

**Table 1 ijms-20-02966-t001:** Data statistics of the small RNA library.

Type	HRK0-1	HRK0-2	HRK48-1	HRK48-2	HSK0-1	HSK0-2	HSK48-1	HSK48-2
Total reads	18187564	15645006	15811473	22213829	15877691	16418090	19154749	19705533
Clean reads	16725511 (91.96%)	14594887 (93.29%)	14335242 (90.66%)	20230796 (91.07%)	14697929 (92.57%)	15060199 (91.73%)	17541500 (91.58%)	18523834 (94.00%)
Mapped reads	15092682 (90.24%)	13055191 (89.45%)	12999357 (90.68%)	18078009 (89.36%)	13090959 (89.07%)	13620528 (90.44%)	15751832 (89.80%)	16892630 (91.19%)
Intergenic	9443266 (56.46%)	7076994 (48.49%)	7502556 (52.34%)	10774382 (53.26%)	8035260 (54.67%)	7314000 (48.57%)	9032395 (51.49%)	9351270 (50.48%)
Exon	2296806 (13.73%)	2735655 (18.74%)	2230122 (15.56%)	3074662 (15.20%)	2018611 (13.73%)	2881758 (19.13%)	2785970 (15.88%)	3185485 (17.20%)
Intron	2197086 (13.14%)	2293967 (15.72%)	2330038 (16.25%)	2958235 (14.62%)	2056682 (13.99%)	2628373 (17.45%)	2986540 (17.03%)	3410640 (18.41%)
miRNA	929540 (5.56%)	607305 (4.16%)	536353 (3.74%)	959353 (4.74%)	748416 (5.09%)	528713 (3.51%)	641115 (3.65%)	584366 (3.15%)
rRNA	211127 (1.26%)	345408 (2.37%)	395420 (2.76%)	295084 (1.46%)	233612 (1.59%)	262057 (1.74%)	294055 (1.68%)	346254 (1.87%)
snoRNA	5719 (0.03%)	6779 (0.05%)	5929 (0.04%)	9232 (0.05%)	4486 (0.03%)	6169 (0.04%)	7500 (0.04%)	8553 (0.05%)
tRNA	1767 (0.01%)	2581 (0.02%)	5226 (0.04%)	6462 (0.03%)	1742 (0.01%)	1970 (0.01%)	1915 (0.07%)	3580 (0.02%)
sncRNA	4662 (0.03%)	5472 (0.04%)	7182 (0.05%)	7141 (0.04%)	5164 (0.04%)	5714 (0.04%)	5719 (0.03%)	5406 (0.03%)
Precursor	30472 (0.18%)	7206 (0.05%)	6290 (0.04%)	34022 (0.17%)	5193 (0.04%)	7113 (0.05%)	2449 (0.01%)	51321 (0.28%)
Unmap	1604212 (9.59%)	1518313 (10.40%)	1312486 (9.17%)	2110698 (10.43%)	1588107 (10.80%)	1423625 (9.45%)	1771428 (10.10%)	1574369 (8.50%)

Note: HRK represented the highly resistant material Gantai-2-2; HSK represented the highly susceptible material Wan82-178; numbers 0 and 48 represented the processing time; and -1 and -2 represented repetitions 1 and 2, respectively.

**Table 2 ijms-20-02966-t002:** MiRNA conservatism grade analysis and statistics.

Type	Highly-Conserved	Moderately-Conserved	Lowly-Conserved	Non-Conserved	Total
No. of known miRNAs	188	23	32	185	428
No. of known miRNAs families	23	3	20	146	192

**Table 3 ijms-20-02966-t003:** Correlation analysis of differentially expressed miRNA and negative regulation of differentially expressed mRNA.

Control Groups	miRNA	miRNA_log2Ratio (Fold Change)	mRNA	mRNA_log2Ratio (Fold Change)	NR Description
HRK48/HRK0	Gma-miR1535a	1.57	Glyma.08G264900.2	−2.25	phototropin-2-like
	Gma-miR166u	−1.70	Glyma.18G204800.1	3.42	ammonium transporter AMT2.2
	Gma-miR394a-3p	−2.26	Glyma.14G004300.1	3.71	nudix hydrolase 2-like
	Gma-miR394a-3p	−2.26	Glyma.05 G073500.2	7.12	F-box protein SKIP14-like
	Gma-miR4996	1.77	Glyma.U019800.1	−7.57	auxin response factor 4-like
	Gma-miR5374-3p	−2.20	Glyma.17G128200.2	7.21	cyclin-dependent kinase G-2-like
	Novel_miR2	1.19	Glyma.05G167800.2	−3.78	guanosine nucleotide diphosphate dissociation inhibitor 2
HSK48/HSK0	Gma-miR1512b	3.10	Glyma.09G019300.1	−6.94	protein FAR1-RELATED SEQUENCE 6
	Gma-miR1535a	2.22	Glyma.19G036600.2	−7.41	probable leucine-rich repeat receptor-like serine/threonine-protein kinase At5g15730
	Gma-miR166j-3p	−1.25	Glyma.07G016700.2	8.54	homeobox-leucine zipper protein ATHB-15-like isoform X4
	Gma-miR166u	−1.42	Glyma.07G016700.2	8.54	homeobox-leucine zipper protein ATHB-15-like isoform X4
	Gma-miR395g	−1.21	Glyma.14G005800.3	8.16	nuclear pore complex protein NUP96
	Gma-miR395g	−1.21	Glyma.14G005800.4	8.10	nuclear pore complex protein NUP96
	Gma-miR399b	−2.31	Glyma.03G021900.2	7.06	uncharacterized protein LOC100306494 isoform X1
	Gma-miR5761a	2.34	Glyma.11G210400.3	−2.68	glycinol 4-dimethylallyltransferase-like
	Gma-miR9749	−1.30	Glyma.09G236600.3	7.32	lipoate-protein ligase LplJ
	Novel_miR2	3.13	Glyma.05G167800.2	−9.51	guanosine nucleotide diphosphate dissociation inhibitor 2
HRK0/HSK0	Gma-miR156q	−1.10	Glyma.06G238100.1	2.37	squamosa promoter-binding protein 1-like
	Gma-miR319d	1.65	Glyma.08G087400.1	−5.22	uncharacterized protein LOC100778760
	Gma-miR394a-3p	1.66	Glyma.05G073500.2	−7.96	F-box protein SKIP14-like
	Gma-miR396e	3.89	Glyma.13G159700.1	−6.85	hypothetical protein GLYMA_13G159700
	Gma-miR4996	−3.14	Glyma.06G040600.1	9.05	THO complex subunit 5A-like isoform X1
	Gma-miR5769	−1.27	Glyma.02G186100.2	8.96	calcium-transporting ATPase 4, plasma membrane-type-like
	Novel_miR36	1.99	Glyma.07G178800.2	−6.80	pentatricopeptide repeat-containing protein At2g15820, chloroplastic-like
HRK48/HSK48	Gma-miR166b	4.63	Glyma.07G016700.2	−8.54	homeobox-leucine zipper protein ATHB-15-like isoform X4
	Gma-miR166j-3p	1.63	Glyma.07G016700.2	−8.54	homeobox-leucine zipper protein ATHB-15-like isoform X4
	Gma-miR395g	1.38	Glyma.14G005800.3	−8.16	nuclear pore complex protein NUP96
	Gma-miR398b	−1.47	Glyma.01G055700.2	3.19	hypothetical protein GLYMA_01G055700
	Gma-miR4416a	−1.15	Glyma.19G136400.2	7.88	putative calcium-transporting ATPase 11, plasma membrane-type
	Gma-miR4996	−1.73	Glyma.06G040600.1	8.90	THO complex subunit 5A-like
	Gma-miR4996	−1.73	Glyma.07G002800.2	7.50	ubiquitin receptor RAD23b-like
	Gma-miR5761a	−1.30	Glyma.13G004400.1	2.45	zinc transporter 8-like
	Novel_miR36	2.66	Glyma.06G088200.3	−7.97	amino acid permease 6-like

Note: HRK represented the highly resistant material Gantai-2-2; HSK represented the highly susceptible material Wan82-178; and the numbers 0 and 48 represented the processing times.

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
