# Peer review of "Determination of the MiRNAs Related to Bean Pyralid Larvae Resistance in Soybean Using Small RNA and Transcriptome Sequencing"

_ijms, 2019, doi:10.3390/ijms20122966_

Round 1
Reviewer 1 Report
Always list scientific names for all plants, insects, etc. A bean pyralid in China may not be the same as a bean pyralid in Brazil.
Introduction is very short and doesn't give the reader the understanding of the importance of the work.
Table 1 and the listings of the number of sequences in the body of the paper is redundant. Remove.
Line 92. Fix.
My major concern is over the two varieties used; Gantai-2-2 and Wan82-178. No information is given in relationship to their breeding and background. How do you know that the differences you found between the two aren't from their different genetic background? This is very important to address and will significantly affect your results.
Figures and tables are too long. Add as supplemental material and focus on the most important aspects.
Why 48 hr after feeding? Upregulation in soybean is highest at 72 hr post insect inoculation as per the literature.
What is the overall value of this project? How will it be used to help soybean producers? Lacking.
Author Response
Point 1: Always list scientific names for all plants, insects, etc. A bean pyralid in China may not be the same as a bean pyralid in Brazil.
Response 1: Thanks for your comment. Bean pyralid is widely distributed throughout the world and is found in Korea, Japan, China, India, Americas, and Africa. We have added the Latin word for bean pyralid (Lamprosema indicata Fabricius) in the article (Line 59-61).
Point 2: Introduction is very short and doesn't give the reader the understanding of the importance of the work.
Response 2: Thank you, your suggestion is good. We have added some contents in the introduction (Line59-78, line85-94).
Point 3: Table 1 and the listings of the number of sequences in the body of the article is redundant. Remove.
Response 3: Thank you, your suggestion is good. We have deleted the number of sequences in the article (Line 100-105).
Point 4: Line 92. Fix.
My major concern is over the two varieties used; Gantai-2-2 and Wan82-178. No information is given in relationship to their breeding and background. How do you know that the differences you found between the two aren't from their different genetic background? This is very important to address and will significantly affect your results.
Response 4: Thanks for your comment. Our project team have carried out a long-term study on soybean resistance to bean pyralid. Gantai-2-2 from Jiangsu was the highly resistant material and Wan82-178 from Anhui was the highly susceptible material, and their resistance to bean pyralid is stable. There are genetic differences between the two breeds (Line70-74). We have compared two resistant materials with different genotypes before and after bean pyralid larvae stress(HSK0/HRK0, HSK48/HRK48), and identified the miRNA/ target genes associated with genotypes, indicating differences in their genetic background (Line160-175).
Point 5: Figures and tables are too long. Add as supplemental material and focus on the most important aspects.
Response 5: Thank you, your suggestion is good. We have deleted Figure 5, and table 3 is modified to supplemental table S1, table 5 is modified to supplemental table S2 in the article.
Point 6: Why 48 hr after feeding? Upregulation in soybean is highest at 72 hr post insect inoculation as per the literature.
Response 6: Thanks for your comment. After some years of resistance identification, the study showed that, when the bean pyralid feeding 48 h, the soybean response time to insects feeding is sufficient, the differential expression of genes is very significant, so this time can evaluate the differences of the highly resistant line and highly susceptible line. When the highly resistant line (Gantai-2-2) and highly susceptible line (Wan82-178) were fed by bean pyralid larvae for 0 h and 48 h, the Illumina HiSeq 2000 platform was used to sequence the transcriptome of soybean leaves infested with bean pyralid larvae.
Point 7: What is the overall value of this project? How will it be used to help soybean producers? Lacking.
Response 7: Thank you, your suggestion is good. We have added some words “about the value of this project and how will it be used to help soybean producers” in the article. (Line 85-94).
Special thanks to you for your good comments.

Reviewer 2 Report
The manuscript by Zeng et al describes changes in soybean miRNA expression in response to bean pyralid larvae feeding. Their analysis, which includes resistant and susceptible soybean lines, identified several miRNAs differentially expressed in response to the herbivore. Target prediction software identified putative targets of these miRNAs, and comparison with a previous transcriptome analysis using the same lines and conditions found soybean mRNAs that are negatively correlated with differentially expressed miRNAs strongly suggesting posttranscriptional regulation.
The experimental setup seems adequate, and the methods are standard and do not present issues. There are, however, some important details missing. The authors also mention the “functional analysis” of miRNA targets, but their approach is only a computational approach that allows the identification of biochemical or molecular pathways that are potentially affected by miRNA regulation and not a truly functional analysis (i.e. genetic and physiological approach such as production of plants with reduced expression of individual miRNAs and evaluation of their defense response) and thus some of the conclusions seem an overreach. Specific points are detailed below:
1. The length distribution of the small RNAs sequenced in this work is not similar to previous published soybean small RNA profiles. In this manuscript the most abundant sRNAs are 20 nt long, while other papers show a very significant peak at 21 nt (see for example Supplementary Figure 1 in [Arikit et al, (2014) Plant Cell 26: 4584–4601]. The authors should explain why their data look different than previous analyses. Is this a technical issue?
2. Line 94: the authors state that the small RNAs identified were miRNA and siRNA but do not explain how they reached this conclusion.
3. Lines 94-95: The authors mention that the miRNAs identified are tissue specific and have conserved sequence, however Figure 1 does not show any of these characteristics. Is this inferred from literature? If so, appropriate references are needed. Otherwise, the phrase should be eliminated.
4. Lines 103-104: The criteria to determine whether the libraries were of good quality are not clear. How did the authors reach the 60% cutoff? Again, if this is a method obtained from literature, it should be cited. Otherwise, explain.
5. Paragraph starting in line 119: The authors used miRNA prediction software to identify new miRNAs, and report that “46 novel miRNAs” were identified. Please add the word “putative” as none of these novel RNAs has been verified.
6. Lines 183-185: The authors state that the differentially expressed miRNAs “defend against the pests”. This is not necessarily true. The results show that the miRNA expression is responsive to insect feeding, but it does not show participation in defense. This could be speculated, but miRNA expression could also be related to physiological changes that result as consequence of insect stress but are not related to defense (such as a general stress response, water stress, etc).
7. Lines 185-186: I do not understand the meaning of this sentence.
8. Figure 2 doesn't look right (problem with the pdf?). I cannot see the dendrogram portion of the heatmap. Also, the clustering seems odd. Visual analysis suggests that lanes 1 and 3 are more similar to each other than to the other two lanes (but I can't see labels or dendrograms, so I am not sure of what is shown).
9. Line 212 (and rest of this section): Please refer to the targets as “predicted” or “putative”, since they have not been experimentally confirmed.
10. Lines 262-265: Again, like in point #6, the results presented in this manuscript show a correlation, not a true functional analysis; thus, the results do not “determined” that the processes identified “play important roles in the defense process”.
11. Figure 5: it is unclear what is shown in this figure. Instead of showing "top 20" (the criteria for "top" are not evident, since there seems to be no relation to significance or number of genes), the figure could show only those categories where statistical significance was observed. Also, what is the "Q value" shown here? Please define. What is the Rich factor depicted? Please define.
12. Section 2.7 (starting on line 272): this section is unnecessarily long. There is no need to repeat all the data presented in Table 4 (and avoid using "and so on"). The summary of how many miRNA/genes in each comparison is enough. The rest doesn't add anything.
13. LOX2 is not a protease. It is a fatty acid lypoxigenase.
14. Lines 383-385: Gma-miRNA319d is up-regulated in the Gantai/Wan comparison at time=0. This would mean that JA signaling is down-regulated in the resistant line. JA is the main hormone controlling defense against herbivores, and the transcriptome analysis previously published by the authors show JA signaling as part of the response to the herbivore in Gantai. These observations should be discussed in more detail.
15. Line 428-429: More information is needed on the test insects. The manuscript uses only common name and this is not enough for identification. Please provide the scientific name, and the source of insects. Were the test insects reared in the laboratory (how?) or were they obtained from the field (in that case, how were they collected? how was identification performed?). What does 4-age mean? Is this the larvae 4th instar?
16. Raw RNAseq data should be deposited in a public database (such as NCBI GEO) as requested by the journal instructions, before submission.
17. qPCR section (starting on line 528): please specify whether these are the same RNA samples used for RNAseq or if they are new RNAs prepared from similar material? That is, is this a technical confirmation or a confirmation using biological replicates?
18. Lines 548-549 (and rest of the section): As mentioned above, this manuscript does not identifies miRNAs that “were confirmed to be related to soybean resistance to bean pyralid larvae”. There is not direct functional analysis of any miRNA and/or target that would “confirm” that any of the miRNAs is directly involved in defense regulation. The manuscript shows an interesting correlation, and some of the genes identified are good candidates for further analysis, but there is no experiment here showing a direct defense function for any miRNA or target gene.
19. Line 559: the authors enumerate some genes that “may also have been related to soybean resistance”. "being related" doesn't really inform the reader. There is more nuance to this interaction. If the miRNA is induced and the target TF (or other) is then repressed, the TF would have a negative effect on resistance, and vice-versa. The analysis overall seems a bit cursory. I suggest that at least in the conclusions the authors should try to group target genes into two classes: those that may have a positive effect on resistance and those that may have a negative effect. In particular, it seems that many of the targets are TF that regulate growth and development, and it is well known that in plants there is an antagonism between growth and defense. Thus, repression of growth may result in increased defenses.
Minor issues:
20. Line 112: The use of the word “conservative” (twice) is incorrect. Please replace in both places with “conserved”.
21. Line 581: Should say “Squamosa”
Author Response
Point 1: The length distribution of the small RNAs sequenced in this work is not similar to previous published soybean small RNA profiles. In this manuscript the most abundant sRNAs are 20 nt long, while other articles show a very significant peak at 21 nt (see for example Supplementary Figure 1 in [Arikit et al, (2014) Plant Cell 26: 4584–4601]. The authors should explain why their data look different than previous analyses. Is this a technical issue?
Response 1: Thanks for your comment. There are two main reasons: 1. All small RNA types, including piRNA, snoRNA, miRNA, etc., were counted here, while previous studies mainly counted the length of miRNA. 2. Previous studies have shown that mRNAs including different tissues and processed expressions, the miRNA have tissue specific and spatio-temporal specificity, which may be related to the collected tissue position. We only collected leaves for sequencing, which cannot represent the whole soybean situation.
Point 2: Line 94: the authors state that the small RNAs identified were miRNA and siRNA but do not explain how they reached this conclusion.
Response 2: Thanks for your comment. We used AASRA to map clean reads to reference genome, and they can be compared as small RNA. Then, bowtie2 was used to compare clean reads with piRNA, snoRNA, miRBase21 and other RNA libraries. Meanwhile, cmsearch was used to compare Rfam, and small RNA was classified according to the comparison results, so as to identify known miRNA and siRNA. Unannotated miRNAs were used to determine new mRNAs based on RIPmiR prediction of whether miRNA precursors were secondary hairpin structures.
Point 3: Lines 94-95: The authors mention that the miRNAs identified are tissue specific and have conserved sequence, however Figure 1 does not show any of these characteristics. Is this inferred from literature? If so, appropriate references are needed. Otherwise, the phrase should be eliminated.
Response 3: Thank you, your suggestion is good. We are very sorry for our incorrect writing, we have deleted “These findings indicated that the miRNA were also tissue specific and sequence conserved.” in the article (Line 113).
Point 4: Lines 103-104: The criteria to determine whether the libraries were of good quality are not clear. How did the authors reach the 60% cutoff? Again, if this is a method obtained from literature, it should be cited. Otherwise, explain.
Response 4: Thank you, your suggestion is good. We have added the literature in the article (Line 122). (17. Ding, X.L. Identification and functional study of differential miRNA in cytoplasmic-nuclear male sterile line NJCMS1A and its maintainer, restorer of soybean.)
Point 5: Paragraph starting in line 119: The authors used miRNA prediction software to identify new miRNAs, and report that “46 novel miRNAs” were identified. Please add the word “putative” as none of these novel RNAs has been verified.
Response 5: Thank you, your suggestion is good. It is really true as your suggestion, we have added “putative” in the line 138.
Point 6: Lines 183-185: The authors state that the differentially expressed miRNAs “defend against the pests”. This is not necessarily true. The results show that the miRNA expression is responsive to insect feeding, but it does not show participation in defense. This could be speculated, but miRNA expression could also be related to physiological modifications that result as consequence of insect stress but are not related to defense (such as a general stress response, water stress, etc).
Response 6: Thanks for your comment. We are very sorry for our incorrect writing, so we have deleted the words “and defend against the pests” in the article (Line263).
Point 7: Lines 185-186: I do not understand the meaning of this sentence.
Response 7: Thanks for your comment. We are very sorry for our negligence, so the statement is incorrect, so we have deleted the words “Also, it was found that the resistant material contained miRNAs which had reacted against the insect pests themselves.” in the article (Line199). Thanks to you for your good comments.
Point 8: Figure 2 doesn't look right (problem with the pdf?). I cannot see the dendrogram portion of the heatmap. Also, the clustering seems odd. Visual analysis suggests that lanes 1 and 3 are more similar to each other than to the other two lanes (but I can't see labels or dendrograms, so I am not sure of what is shown).
Response 8: Thanks for your comment. We are very sorry for our negligence, the figure 2 we provided was incorrect, so we have now put the correct figure 2 into the article. I’m sorry.
Point 9: Line 212 (and rest of this section): Please refer to the targets as “predicted”or “putative”, since they have not been experimentally confirmed.
Response 9: Thank you, your suggestion is good. We have added “predicted” in the line225, line227 and line229.
Point 10: Lines 262-265: Again, like in point #6, the results presented in this manuscript show a correlation, not a true functional analysis; thus, the results do not “determined” that the processes identified “play important roles in the defense process”.
Response 10: Thank you, your suggestion is good. We have modified "determined" to “speculated” in the line276, line280. Special thanks to you for your good comments.
Point 11: Figure 5: it is unclear what is shown in this figure. Instead of showing "top 20" (the criteria for "top" are not evident, since there seems to be no relation to significance or number of genes), the figure could show only those categories where statistical significance was observed. Also, what is the "Q value" shown here? Please define. What is the Rich factor depicted? Please define.
Response 11: Thanks for your comment. RichFactor is the ratio of DESs target genes numbers annoted in this pathway term to all gene numbers annoted in this pathway term. Greater richFator means greater intensiveness. Qvalue is corrected pvalue ranging from 0~1, and less Qvalue means greater intensiveness. We just display the top 20 of enriched pathway terms.
Point 12: Section 2.7 (starting on line 272): this section is unnecessarily long. There is no need to repeat all the data presented in Table 4 (and avoid using "and so on"). The summary of how many miRNA/genes in each comparison is enough. The rest doesn't add anything.
Response 12: Thank you, your suggestion is good. We have deleted some contents in the Section 2.7.
Point 13: LOX2 is not a protease. It is a fatty acid lypoxigenase.
Response 13: Thank you, your suggestion is good. We are very sorry for our negligence, we have deleted “protease” in the line373.
Point 14: Lines 383-385: Gma-miRNA319d is up-regulated in the Gantai/Wan comparison at time=0. This would mean that JA signaling is down-regulated in the resistant line. JA is the main hormone controlling defense against herbivores, and the transcriptome analysis previously published by the authors show JA signaling as part of the response to the herbivore in Gantai. These observations should be discussed in more detail.
Response 14: Thank you, your suggestion is good. We have added some contents in the article (Line380-383, line387-391).
Point 15: Line 428-429: More information is needed on the test insects. The manuscript uses only common name and this is not enough for identification. Please provide the scientific name, and the source of insects. Were the test insects reared in the laboratory (how?) or were they obtained from the field (in that case, how were they collected? how was identification performed?). What does 4-age mean? Is this the larvae 4th instar?
Response 15: Thank you, your suggestion is good. The test insect is bean pyralid (Lamprosema indicata Fabricius), they obtained from the field, and we asked the insect experts to help identify insect age. The “4-age” mean “larvae 4th instar”. We have made modifications in the article (Line431-432).
Point 16: Raw RNAseq data should be deposited in a public database (such as NCBI GEO) as requested by the journal instructions, before submission.
Response 16: Thank you, your suggestion is good. All data were submitted to the National Center for Biotechnology Information (NCBI) under SRA number SRR7903818, SRR7903819, SRR7903820, SRR7903821, SRR7903822, SRR7903823, SRR7903824, SRR7903825.
Point 17: qPCR section (starting on line 528): please specify whether these are the same RNA samples used for RNAseq or if they are new RNAs prepared from similar material? That is, is this a technical confirmation or a confirmation using biological replicates?
Response 17: Thanks for your comment. qPCR and RNAseq used the same RNA sample.
Point 18: Lines 548-549 (and rest of the section): As mentioned above, this manuscript does not identifies miRNAs that “were confirmed to be related to soybean resistance to bean pyralid larvae”. There is not direct functional analysis of any miRNA and/or target that would “confirm” that any of the miRNAs is directly involved in defense regulation. The manuscript shows an interesting correlation, and some of the genes identified are good candidates for further analysis, but there is no experiment here showing a direct defense function for any miRNA or target gene.
Response 18: Thank you, your suggestion is good. We have modified “revealed” to “speculated”, the statements of “the function analysis, target genes” were corrected as “the pathway analysis, predicted target genes” in the line 552.
Point 19: Line 559: the authors enumerate some genes that “may also have been related to soybean resistance”. "being related" doesn't really inform the reader. There is more nuance to this interaction. If the miRNA is induced and the target TF (or other) is then repressed, the TF would have a negative effect on resistance, and vice-versa. The analysis overall seems a bit cursory. I suggest that at least in the conclusions the authors should try to group target genes into two classes: those that may have a positive effect on resistance and those that may have a negative effect. In particular, it seems that many of the targets are TF that regulate growth and development, and it is well known that in plants there is an antagonism between growth and defense. Thus, repression of growth may result in increased defenses.
Response 19: Thank you, your suggestion is good. We have made some modifications in the line558-562
Minor issues:
Point 20: Line 112: The use of the word “conservative” (twice) is incorrect. Please replace in both places with “conserved”.
Response 20: Thank you. We have modified “conservative” to “conserved” in the line131.
Point 21: Line 581: Should say “Squamosa”
Response 21: Thank you.We have modificationd “Quamosa” to “Squamosa” in the article (Line586).
Once again, thank you very much for your comments and suggestions.
Round 2
Reviewer 1 Report
Minor English editing needed.
Author Response
Point 1: Comments and Suggestions for Authors Minor English editing needed.
Response 1: Thanks for your comment and suggestion. We have maded
some modifications in the article.